# Study of Global Navigation Satellite System Receivers’ Accuracy for Unmanned Vehicles

**DOI:** 10.3390/s24185909

**Published:** 2024-09-12

**Authors:** Rosen Miletiev, Peter Z. Petkov, Rumen Yordanov, Tihomir Brusev

**Affiliations:** 1Faculty of Telecommunication, Technical University of Sofia, 1000 Sofia, Bulgaria; miletiev@tu-sofia.bg (R.M.); brusev@ecad.tu-sofia.bg (T.B.); 2Faculty of Electronics, Technical University of Sofia, 1000 Sofia, Bulgaria; yordanov@tu-sofia.bg

**Keywords:** GNSS, patch antenna, helix antenna, multi-constellation reception

## Abstract

The development of unmanned ground vehicles and unmanned aerial vehicles requires high-precision navigation due to the autonomous motion and higher traffic intensity. The existing L1 band GNSS receivers are a good and cheap decision for smartphones, vehicle navigation, fleet management systems, etc., but their accuracy is not good enough for many civilian purposes. At the same time, real-time kinematic (RTK) navigation allows for position precision in a sub-centimeter range, but the system cost significantly narrows this navigation to a very limited area of applications, such as geodesy. A practical solution includes the integration of dual-band GNSS receivers and inertial sensors to solve high-precision navigation tasks, but GNSS position accuracy may significantly affect IMU performance due to having a great impact on Kalman filter performance in unmanned vehicles. The estimation of dilution-of-precision (DOP) parameters is essential for the filter performance as the optimality of the estimation in the filter is closely connected to the quality of a priori information about the noise covariance matrix and measurement noise covariance. In this regard, the current paper analyzes the DOP parameters of the latest generation dual-band GNSS receivers and compares the results with the L1 ones. The study was accomplished using two types of antennas—L1/L5 band patch and wideband helix antennas, which were designed and assembled by the authors. In addition, the study is extended with a comparison of GNSS receivers from different generations but sold on the market by one of the world’s leading GNSS manufacturers. The analyses of dilution-of-precision (DOP) parameters show that the introduction of dual-band receivers may significantly increase the navigation precision in a sub-meter range, in addition to multi-constellation signal reception. The fast advances in the performance of the integrated CPU in GNSS receivers allow the number of correlations and tracking satellites to be increased from 8–10 to 24–30, which also significantly improves the position accuracy even of L1-band receivers.

## 1. Introduction

The contemporary world needs more and more precise navigation. The main functions of Global Navigation Satellite Systems (GNSS) are to provide the moving object geographic coordinates, the parameters of the speed vector and accurate time. The accuracy of these parameters depends on lots of other parameters, such as the number of satellite signals available, system integrity, satellite positions, etc. One of the possible ways to improve position accuracy is multi-GNSS systems, which have become more common owing to an increase in the number of satellite systems [1,2,3,4] as the spatial geometric distribution of GPS satellites has been significantly improved after the addition of BDS satellites [5]. 

Two GNSS services are offered with different accuracy and levels of access: standard accuracy service and precise service. These services are described in the Standard Positioning Service Performance Standard (SPS) [6], and the Precise Point Positioning Standard (PPP) [7]. In addition, the position accuracy of both services, considering a given probability and time accuracy, is provided with specific values on the European Space Agency website.

The standard accuracy position and time service are provided by the C/A (coarse/acquisition) ranging and acquisition code and the navigation message, which modulate the L1 carrier frequency and are intended for civilian users. The standard accuracy for position and time is determined based on a probability of 95% and only up to the signal-in-space (SIS) level. The accuracy improvement requires methods based on additional technical means [8,9]. In some applications, such as unmanned aerial vehicles [10], transport infrastructure measurements [11], unmanned ground vehicles [12], navigation systems for measurements [13], etc., the accuracy requirements are higher since sub-meter or centimeter-level precision is critical. Through the C/A code, a standard service (SPS—Standard Positioning Service) is offered with an accuracy of 100 m horizontally, 156 m vertically and with a maximum inaccuracy of the receiver clock of 340 ns. The speed measurement error is 0.3 m/s. The data are at the 95% probability level at the navigation receiver input.

The precision service is delivered by the P (Precise) code and the navigation message, which modulate the L1 and L2 carrier frequencies. Due to the fact that the code is spread over two frequencies, this makes it more robust than the C/A code. Also, ionosphere errors are overcome with both frequencies. The code itself is longer and this contributes to higher accuracy. The P code is a very long (267 days) 10.23 MHz PRN (Pseudo-Random Noise) code, with each satellite transmitting a weekly portion of the 267-day PRN sequence, restarting the code sequence at the end of each week. This code was later replaced by the cryptographically more secure Y code, which is a sum modulo two between the P code and an encryption W code, which is clocked at 0.511 MHz. The P code, as well as its Y variant, are intended for military purposes and provide the PPS (Precise Positioning Service), namely, 22 m horizontally and 27.7 m vertically. The receiver clock does not deviate more than 20 ns from the system time. Again, data are given at the 95% probability level at the navigation receiver input.

Since 2018, the new satellites also broadcast a new civil code, L1C, which should ensure the interoperability of the system with other global navigation systems. To avoid interference with other signals located on the same channel, Multiplexed Binary Offset Carrier (MBOC) modulation is applied to the L1C. L5, which is another civilian signal whose function is to provide safe transportation and other high-performance applications. It is broadcast in a radio frequency band (1176 MHz) which is intended for services related to the safety of air transport. In combination with the L1C/A signal, it enables the elimination of ionospheric errors and higher reliability in determining the position.

The use of the frequency resource by the codes is presented in Figure 1. The PRN codes’ power spectrum is shown in Figure 2 and their parameters are described in Table 1, which should be taken into account in the antenna design.

The addition of new navigation codes leads to a reduction in position errors. The application of dual-band GNSS signal reception is a significant technological step towards next-level accuracy and reliability in location tracking, as the standard point positioning service error is higher than 2.5 m and this is insufficiently accurate for intelligent transport systems [17,18] as the standard lane width is equal to 3 m. Dual-band reception significantly increases position accuracy to the sub-meter level with the ability to correct ionospheric distortions, but even this technology cannot overcome the main disadvantage of global positioning systems, which is defined by the low update rate of up to 10 Hz. In this case, the integration of a dual-band GNSS receiver and Inertial Measurement Unit (IMU) could cover the land vehicle requirements, as the GNSS system could eliminate IMU error accumulation [19,20] while the IMU system could increase the position update rate up to 1 kHz.

GNSS system accuracy is critical for IMU/GNSS systems as most of the time the position and velocity, as measurement inputs, are used to compensate for IMU errors [21]. The covariance matrix from GNSS data processing is also passed to the fusion algorithm [22]. If it is suggested that the update rate of a GNSS system is 1 Hz and the latest generation accelerometer zero-rate offset is equal to 20 mg [23], the accumulated IMU displacement error Δ for this period of time is equal to 10 cm according to Equation (1) [24].
(1)$$\Delta =\int \int_{0}^{t}adt^{2}$$

Using the same formula, the accumulated IMU error will increase to 1 m for 10 s. 

As the GNSS-only error is higher than the fusion of GNSS and IMU data for navigation [25], especially in urban environments due to multi-path signal interference and signal loss, it is essential to increase the GNSS accuracy and update rate as the long GNSS gaps reduce positioning accuracy distinctly [26]. 

There are multiple benefits of Kalman filters in UAV applications, such as position, altitude and velocity estimation, GNSS and IMU data fusion, and smoothing the sensor output orientation in the space, in terms of roll, pitch and yaw gyro-stabilized platforms in the navigation system, etc. Previous researchers [27] found that the accuracy of a Kalman filter used in an INS/GPS integration algorithm is increased by estimating the measurement covariance matrix, based on measurement data from GPS and the DOP parameters, which are functions of the diagonal elements of the covariance matrix of the parameters, expressed in a global or a local geodetic frame. 

As position accuracy mainly depends on the number of processed satellite signals, the number of satellite systems used, the number of frequency bands used (especially in urban environments) and antenna parameters, the current study will investigate the influence of different parameters on the localization by analyzing the dilution-of-precision (DOP) parameters. The comparison will be made for the following cases:Reception of satellite signals from a single GNSS system.Reception of satellite signals from multiple GNSS systems.Analysis of DOP parameters of L1 and L1/L5 GNSS receivers.Analysis of DOP parameters of each GNSS receiver with active patch and passive helical antennas.Comparison of DOP parameters for GNSS receivers from one manufacturer, but from a different production generation, to evaluate the number of tracking satellites during the time.

## 2. Materials and Methods

### 2.1. System Design

The study of navigation receiver performance was accomplished by preparing a PCB board using the widely used GNSS receivers, such as the NEO or MAX receiver series from U-blox AG company. Pin-compatible receivers have also been introduced by other companies, such as MinewSemi Co., Ltd. (Shenzhen, China). The main positioning task was solved by the GNSS receiver (Figure 3). It is based on different GNSS modules depending on the required accuracy. If standard accuracy is required (σ ≤ 2.5 m), then modules like U-blox NEO-M9N or GP-01 GPS + BDS Compass ATGM332D may be used. If high-precision accuracy is required (sub-meter level), then a MinewSemi ME32GR01 module may be used, which is a multi-constellation (GPS, Beidou, GLONASS and Galileo), concurrent, simultaneous multi-constellation L1 + L5 positioning GNSS module. All the GNSS receivers mentioned are pin-compatible and support UART and I^2^C protocols, while U-blox modules additionally provide a USB connection, which allows high-speed data transmission or direct connection to a PC. As the board is powered by providing 5 V at Vin pin, a CMOS low-dropout (LDO) voltage regulator should be used, such as MCP1700T-3302, to convert the input voltage to 3.3 V as a GNSS-receiver power supply. The selection of this LDO regulator is set due to the very low drop-out voltage of only 178 mV@250 mA load; the LDO output is stable only when using 1 μF output capacitance. Simultaneously, the VDD_USB input at U-blox receivers is powered using another CMOS LDO, such as MCP1824, which is distinguished by enabling (EN) input. 

### 2.2. Helix Antenna Design

For the purpose of this study, a passive helix antenna has been selected, due to its good polarization properties and simplicity of construction. A similar approach in an identical study is considered by [28]. The helix antenna consists of a spiral of wire fed by a coaxial line. The inner conductor of this line is connected to the wound conductor (through a matching section) and the shield to a metallic disk (reflector). The purpose of the reflector is to direct the radiation into one hemisphere and increase the gain, to reduce currents on the outer conductor of the coaxial line and to reduce input impedance fluctuations. The diameter of the disk is typically chosen in the range of order of (0.7 to 2) times diameter of the helix. Some design highlights and relations can be found in [29]. Helix antenna can operate in axial or normal mode of radiation. The former is present when the length of the wire loops is approximately similar to the wavelength and the pitch angle is between 10 and 15 deg. This creates the phase condition for the summation of radiated waves in a forward direction. The later mode is present when the loop length is much less than the wavelength. Typical applications of this type of helix antenna are handheld devices, wireless routers, etc. 

The design considerations for axial-mode, uniform-diameter antenna are led by the requirements of reasonable gain (at least several decibels) and minimal side lobes, while keeping the return loss within reasonable limits. Since only a few parameters can be modified—the helix diameter, pitch and overall length—it is possible to achieve a limited control over these antenna parameters. By changing the pitch and diameter, the propagation constant of the propagating wave, the phase angle of the current at each turn can be controlled. Since the helical antenna can be regarded as an end-fire antenna array, the same design and pattern synthesis principles are applicable, but with a much lesser degree of freedom. The optimization is performed with a 4NEC2 Method of Moments simulator (open source) and confirmed with Ansys HFSS ^®^ (FEM). An analytical solution is proposed in [30]; however, due to the complexity of the problem, the use of electromagnetic simulators is highly recommended. 

In general, there are multiple requirements for GNSS UAV antennas, divided into three major groups: mechanical, electrical and procurement. In relation to the first group: low profile (good aerodynamics), low mass, predictable center of gravity, low complexity of the design and rigid construction. Related to the second one: broad main lobe, phase center position stability, minimal side lobes (to reduce noise and interference), positive realized gain, and well-defined and stable Rx frequency bands in terms of return loss. Related to the third one: low cost of production and capability for mass production. Since the popular ceramic patch antennas often deviate in their electrical performance, an additional helix antenna, as a reference, is considered in this study. 

Assuming the above, an axial mode helix antenna was designed and fabricated. Design properties and dimensions are as follows (in mm) (Figure 4): ground plate disk diameter: 300 mm, number of turns: 6, helix diameter: 65 mm, helix spacing (pitch): 47 mm, wire diameter: 2.3 mm (4 mm^2^ wire gage). Figure 5 and Figure 6 represent the antenna gain of the antenna for 1.2 GHz and 1.575 GHz, respectively. The achieved gain is 11.3 dBic and 12.5 dBic. 

The cross-polarization properties are presented in Figure 7. It is evident that within the main beam, the XPD varies between −20 dB for the boresight to −15 dB in a cone of ±30 deg. Although cross-polarization discrimination is not critical for GNSS applications, for the purpose of the study and methodology purity, such a high value is desirable. 

The matching of the helix antenna is performed by bending the conductor that is closest to the connector toward the ground plane. This changes the characteristic impedance of the wire and creates a quarter-wave transformer, which matches the helix impedance (120–140 Ohm) to one of the connectors (75 Ohm). The simulated results of the return loss are presented in Figure 8 and show acceptable performance for a receive-only antenna. The phase center position estimation for both frequency bands is presented on Figure 9 and Figure 10, showing stable performance at approx. 20 mm from the ground plane. 

Therefore, the developed helix antenna may successfully serve the purpose of a receive device in the meter positioning range as it covers the GNSS frequency bands L1, L5 and L2.

The results show that the maximum horizontal component error of the phase center is about 1.4 mm and the vertical component error is about 2.6 mm for GPS receivers’ antennas of the Trimble, Leica and Ashtech GPS receivers [31]. For different types of antennas, such as “Trimble to Ashtech”, the maximum horizontal component error is about 4.3 mm and the maximum vertical component error is about 33.5 mm. As for “Trimble to Leica”, the maximum horizontal component error is about 5.2 mm and the maximum vertical component error is about 55 mm. The study of three low-cost GNSS antennas against three high-grade antennas [32] reports noticeable differences in positioning performance with different low-cost antennas. Such an effect may be overcome when using signals from multiple satellite systems. 

## 3. Results

The position-accuracy navigation receivers from different producers were studied using two types of antennas. The first type was a commercial L1/L5 active patch antenna (Figure 11) with the following characteristics:Ceramic size: upper patch—20 × 20 mm, bottom patch—35 × 35 mm.Frequencies: GPS (L1/L5), Galileo (E1), GLONASS (L1), Beidou (B1).Gain: L1: 5 ± 1 dBi, L5: 3 ± 1 dBi.Bandwidth: min 10 MHz.Voltage Standing Wave Ratio (VSWR): ≤2.0.Low-noise amplifier (LNA): Gain (L1: 32 ± 3 dB, L5: 28 ± 3 dB); Noise Figure NF < 1.0 dB; VSWR < 2.0.

The second antenna type was a passive helix wideband antenna, described in Section 2.2. Both antennas were tested simultaneously in outdoor situations. The helix antenna was directed at 45° in azimuth.

The experiments were accomplished using a GNSS software application U-center 2, ver. 24.07.111267 from U-blox AG company (Thalwil, Switzerland). The dilution-of-precision (DOP) parameter was used for measuring the precision of the GNSS receiver position. The horizontal dilution of precision (HDOP) and Position (3D) dilution of precision (PDOP) were calculated for each receiver with ceramic and helix antennas. This parameter was chosen to achieve an improved navigation performance in urban areas. A new GNSS update strategy is proposed in a loosely coupled GNSS/IMU fusion scheme based on the average of the predicted pseudo-range errors and the value of Position Dilution of Precision (PDOP) [33].

For the calculation of the DOP parameters, the almanac data may be used and thus determined before the actual measurement. This parameter is a function of visible satellites and time. The smaller its value, the better the placement of the visible satellites. It is considered to have good geometry if the PDOP has a value of less than three and HDOP has a value of less than two. The best results are obtained when the HDOP value varies from 0.5 to 1.0. With this parameter, a preliminary choice can be made about which satellites will be used when solving the navigation task. It should be noted that this is the parameter that transforms measurement errors into positioning errors.

Here, random errors have been separated from those due to other influences. The total error from the various influences is a sum of the errors of the individual influences, while the random errors (assuming they are all root mean square) combine as the square root of the sum of their squares. To determine the total error, it is suggested in the literature to add the triple total random error to the total influence error. With this tripling, the root mean square error becomes absolute with a confidence probability of 99.7% (corresponds to 3σ).

The experimental study of a high-accuracy GNSS module covers the determination of positioning accuracy in outdoor situations using abovementioned U-center 2 software on the following modules:NEO-F10T (from U-blox AG company)—L1/L5 bands GNSS receiver.ME32GR01 (from MinewSemi Co., Ltd.)—L1/L5 bands GNSS receiver.ATGM332D (from Zhongkewei Group)—L1 band GNSS receiver.

The results obtained from the experimental study of L1/L5 GNSS receivers were compared with the latest generation L1 band receivers, such as the U-blox MAX-M10S, MinewSemi MS32SN1 and Zhongkewei Group ATGM336H modules. The measurement results for each GNSS receiver are shown in Figure 12, Figure 13 and Figure 14, respectively.

The experimental study of GNSS receivers was also extended with a comparison of the receivers from one manufacturer but from a different generation. To achieve this goal, the results obtained from the MAX M10S receiver were compared with the performance of the MAX M8Q receiver, which is shown in Figure 15.

## 4. Discussion

The experimental results show that both antennas may receive GNSS signals in the L1 and L5 frequency bands. Due to the narrow radiation pattern, the reception of the rising and setting satellites is labored and their signal levels are lower compared with the ceramic patch antenna. However, the higher gain of the helix antenna may increase the satellite signal Carrier-to-Noise (C/No) levels over 50 dB/Hz.

Figure 16 represents the results of the tracked satellites of the MinewSemi ME32GR01 receiver. The number of visible satellites using patch and helix antennas are approximately equal (18 vs. 19 satellites for patch and helix antennas, respectively), and the best obtained values for PDOP and HDOP parameters are 1.0 and 0.6, respectively, as multiple GNSS systems are used for localization (GPS, GLONASS and Galileo). It also shows the satellite signal levels for the L1 and L5 bands, which allow the position errors to be reduced to 0.5 m. The study results may be summarized as follows:NEO-F10T (from U-blox AG company)—determination of PDOP and HDOP coefficients. Best results obtained: PDOP = 1.0; HDOP = 0.7.ME32GR01 (from MinewSemi Co., Ltd.)—determination of PDOP and HDOP coefficients. Best results obtained: PDOP = 1.0; HDOP = 0.6.ATGM332D (from Zhongkewei Group)—determination of PDOP and HDOP coefficients. Best results obtained: PDOP = 1.6; HDOP = 1.4.

Similar results were obtained for the latest generation GNSS receiver, U-blox NEO F10T (Figure 17). The number of tracked satellites increased to 24 ÷ 28, but the PDOP and HDOP parameters were not improved compared to the ME32GR01 (MinewSemi, Shenzhen, China) receiver, because L5 satellites are not an appropriate choice for the constellation (the only visible L5 satellite, E25, is not tracked). The reception of L5 signals enhanced the GPS system’s penetration due to its higher power and unique wavelength properties, which offer slightly better penetration capabilities through obstacles like foliage and building materials.

The experimental study of the position accuracy of the latest generation L1 receivers, such as the U-blox MAX-M10S, MinewSemi MS32SN1 and ATGM336H (Zhongkewei Group, Shenzhen, China) modules, which are shown in Figure 12, Figure 13 and Figure 14, respectively, shows the improved position accuracy when the number of constellations used is increased from one (MinewSemi MS32SN1) to four (U-blox MAX-M10S), because the number of tracking satellites could not exceed ten. The best obtained HDOP and PDOP values are [0.9, 1.0] and [1.3, 1.4], respectively.

The next study was a comparison of DOP parameters for GNSS receivers from one manufacturer but from a different production generation. The study covers the PDOP and HDOP measurements of 8th- and 10th-generation MAX-series GNSS receivers from U-blox AG company. The results are presented in Figure 15 and Figure 12, respectively. The significant improvement of DOP parameters of the 10th-generation GNSS receiver (HDOP decreases from 1.10 to 0.68) is explained by the increased number of tracked satellites and the choice of a better satellite constellation.

The experimental study of L1/L5 GNSS receivers shows that the tracking of more than 16–20 satellites leads to excellent position accuracy in the sub-meter range. The receiver deviation maps show accuracy down to 0.5 m when HDOP values are down to 0.6 ÷ 0.7. 

The comparison of the obtained data shows some conclusions:The number of processing satellites is approximately equal for GNSS modules with active patch and passive helix antenna. The wider radiation pattern of the patch antenna is compensated for by the higher gain of the helix antenna. In the multi-constellation cases, where the number of tracked satellites exceeded 16, it is well observed that lots of satellites are not used in navigation or are not tracked.If the GNSS receiver used only one constellation (for example, MS32SN1), then the number of processed satellite signals did not exceed 10. This significantly increases the HDOP values to [0.9, 1.0] and PDOP values to [1.3, 1.4]. The same situation appears if the GNSS receiver uses only a fraction of the received signals to calculate the position, as in the case of the MAX M8Q module and multi-constellation reception of GPS and GLONASS signals.The comparison of the results from the MAX M8Q and MAX M10S receivers shows significant improvement in the number of used constellations from 2 to 4 and the number of the processed satellite signals from 8 ÷ 10 to 16 ÷ 20, which reduced the HDOP value down to 0.6.

The comparison of the position accuracy of L1/L5 GNSS receivers in our previous research [34] shows that the obtained HDOP and PDOP values are improved compared with other GNSS receivers, such as Quectel LC79D and LC29. The significant improvement of dilution-of-precision parameters is also obtained according to other publications [35,36], where the best PDOP values reported were from 1.8 to 2.0 for L1 and L1/L5 GNSS receivers, but the number of processed satellites was limited to 10 [V]. Nevertheless, although the results were measured in Eastern Europe (in Sofia, Bulgaria), similar PDOP values are obtained for the polar regions [37], where minimum, mean and maximum PDOP values for all GNSS satellites are reported as 0.7, less than 1.2 and less than 1.8, respectively.

Future research is to be directed to the study of the position accuracy of IMU/GNSS systems in urban environments where the presence of buildings and trees induce signal reflections and attenuations when the number of visible satellites is not sufficient and GNSS receivers are unable to achieve the required accuracy of unmanned vehicles. The practical realization of such integrated IMU/GNSS systems requires data fusion from both navigation systems, which is accomplished by the Kalman filter because its optimality in fusing data has been proved. 

## 5. Conclusions

High-precision GNSS navigation is introduced more frequently in the civilian life due to the higher requirements for position accuracy. Nevertheless, of the achieved accuracy in sub-meter or sub-centimeter range, GNSS receivers still cannot send position values with update rates which exceed 10 Hz due to the very complex computation task of the correlation processes. Also, GNSS signals may be completely lost in lots of situations, such as tunnels, underground garages, etc. In these cases, autonomous vehicles cannot calculate their position. The integration of GNSS receivers with an inertial measurement unit (IMU) may overcome this navigation problem for a limited period of time, as IMU systems may calculate the spatial object position and its speed and movement direction. However, the position calculation is a relative process, which leads to unlimited error accumulation, but the position refresh rate may exceed 400 Hz. The integration of both navigation systems may significantly reduce their drawbacks. 

The possible applications of the current study may be directed toward intelligent transport systems to evaluate the position accuracy of standalone GNSS receivers or to integrated IMU/GNSS systems to evaluate the parameters of the Kalman filter in unmanned vehicles. The current research results may be used to solve important problems when designing different types of Kalman filter, such as incomplete a priori knowledge of the process noise covariance matrix and measurement noise covariance.

## Figures and Tables

**Figure 1 sensors-24-05909-f001:**
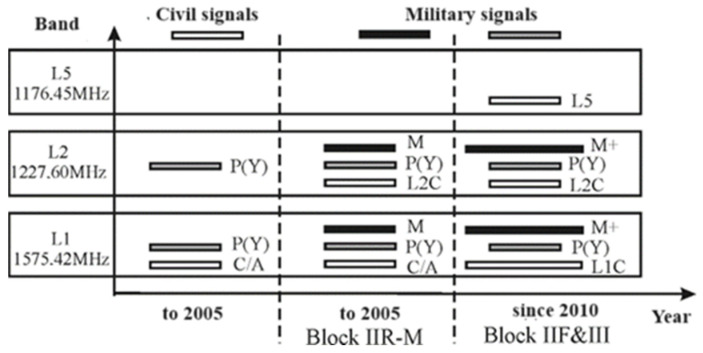
PRN codes in GPS system.

**Figure 2 sensors-24-05909-f002:**
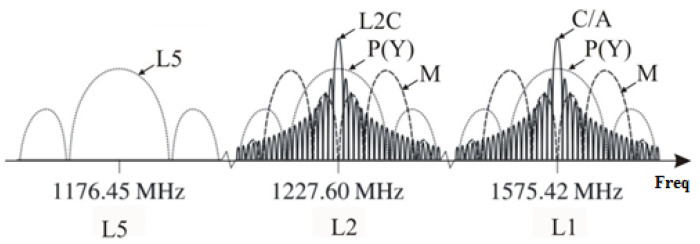
Power spectrum of GPS codes.

**Figure 3 sensors-24-05909-f003:**
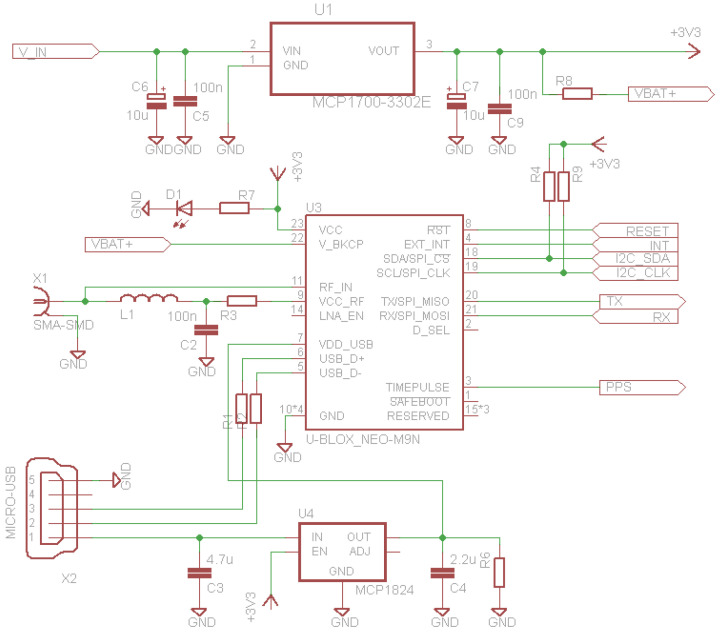
Schematic of the system design.

**Figure 4 sensors-24-05909-f004:**
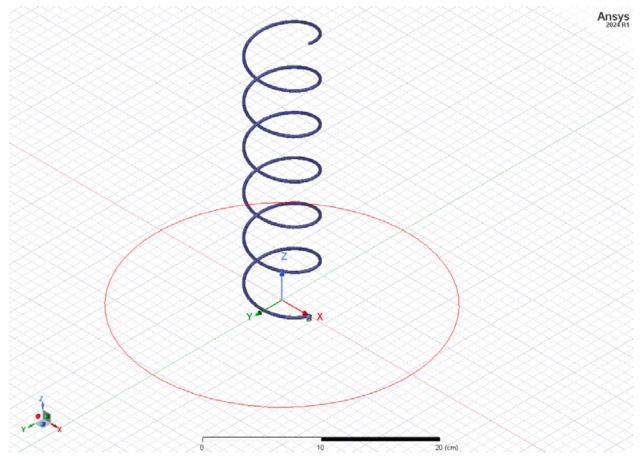
Outline mechanical design of the GNSS helix antenna.

**Figure 5 sensors-24-05909-f005:**
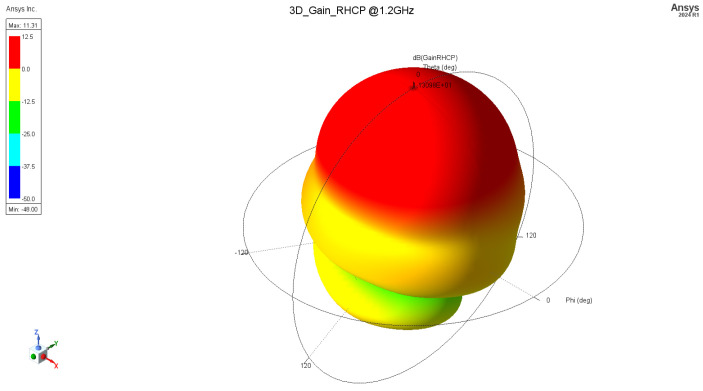
Three-dimensional radiation pattern, RHCP @ 1.2 GHz.

**Figure 6 sensors-24-05909-f006:**
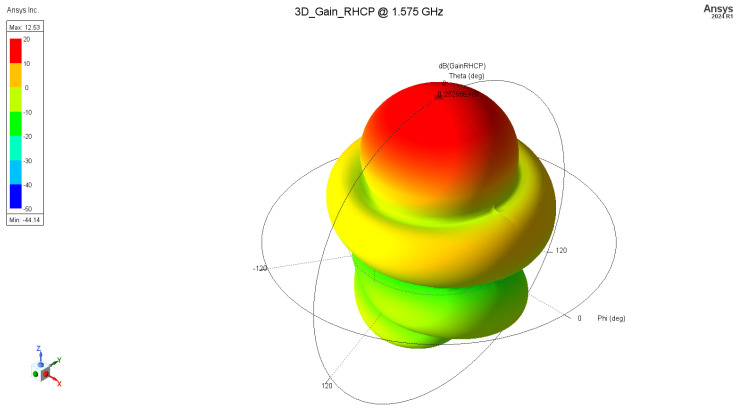
Three-dimensional radiation pattern, RHCP @ 1.575 GHz.

**Figure 7 sensors-24-05909-f007:**
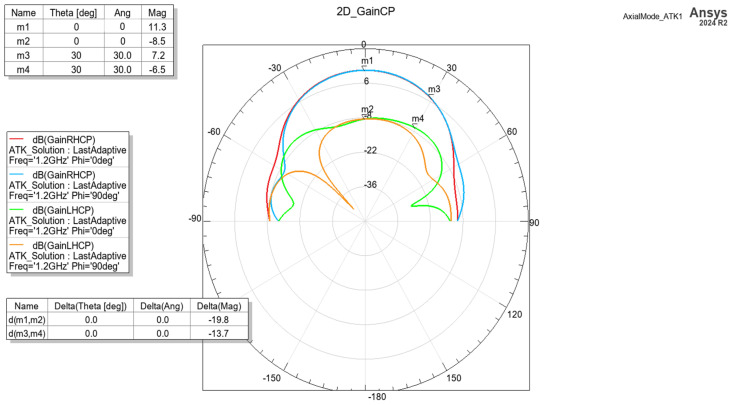
Cross-polar properties of the helix antenna.

**Figure 8 sensors-24-05909-f008:**
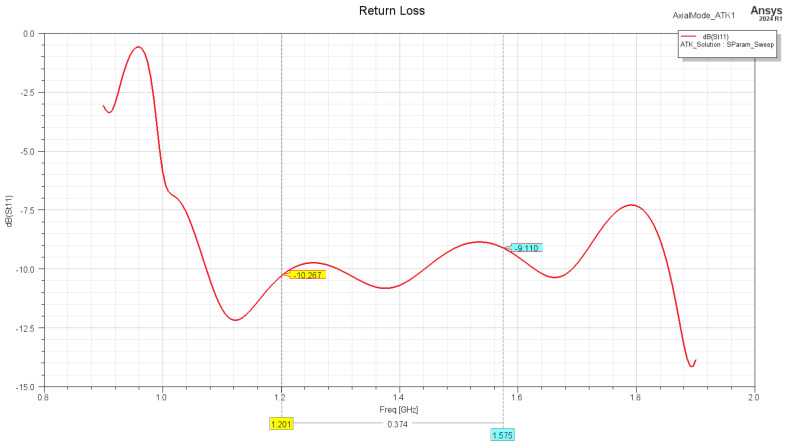
Return loss of the antenna, matched to 75 Ohm port.

**Figure 9 sensors-24-05909-f009:**
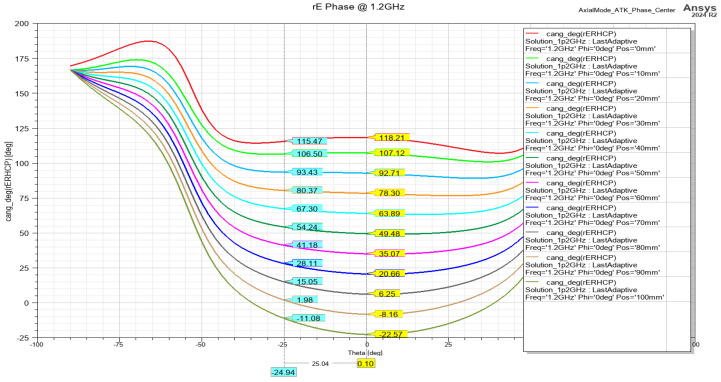
Phase center position estimation from the ground plane @ 1.2 GHz.

**Figure 10 sensors-24-05909-f010:**
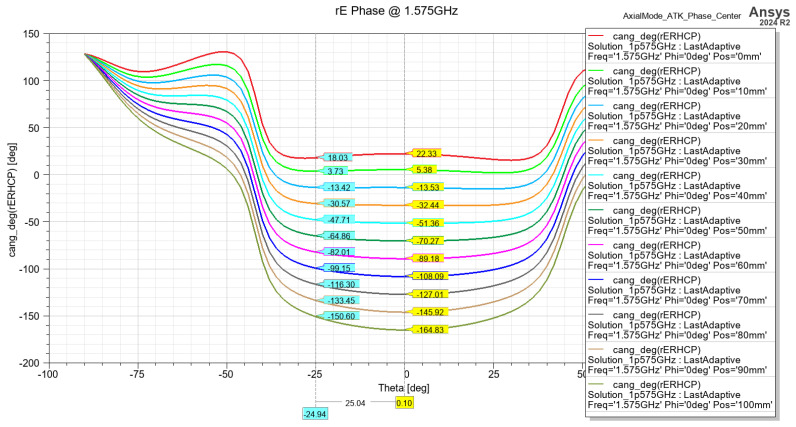
Phase center position estimation from the ground plane @ 1.575 GHz.

**Figure 11 sensors-24-05909-f011:**
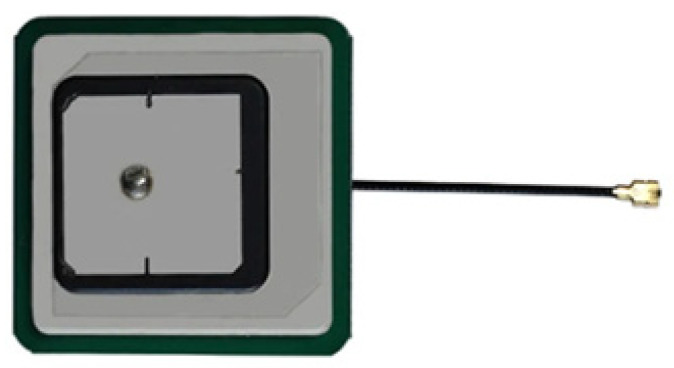
Ceramic patch GNSS L1/L5 antenna.

**Figure 12 sensors-24-05909-f012:**
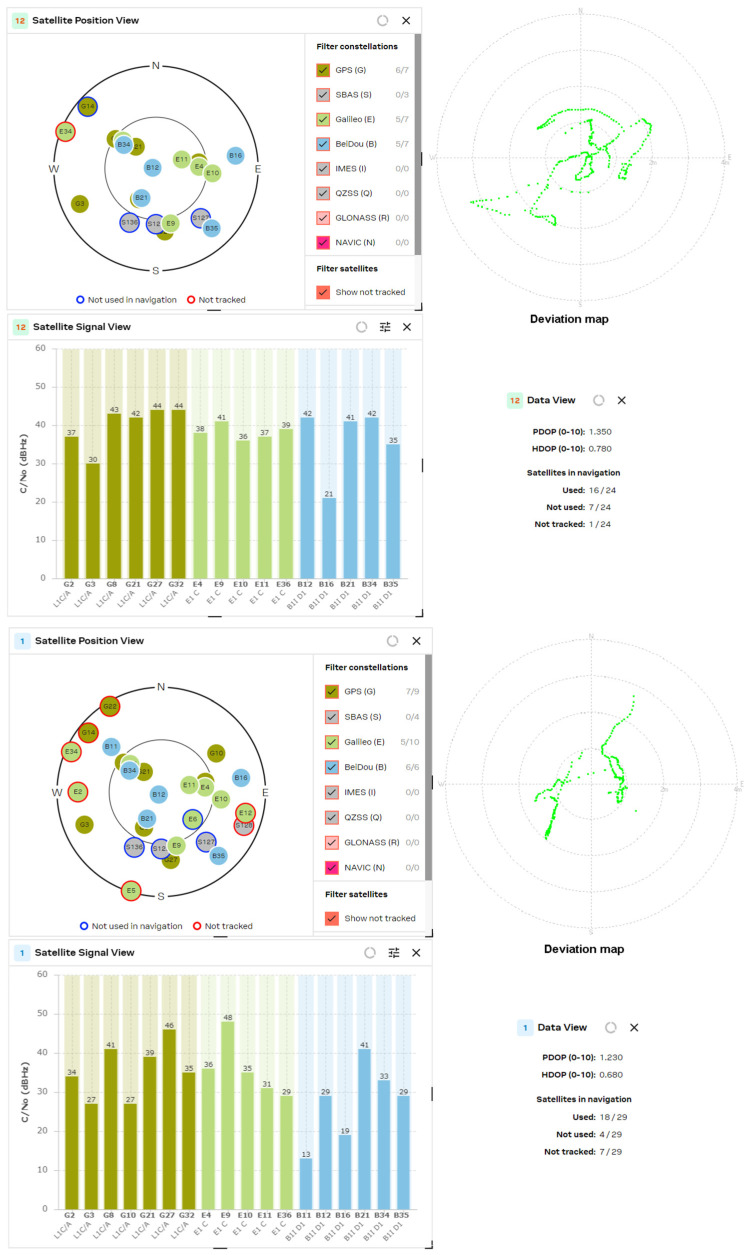
U-blox MAX M10S (patch and helix antenna, respectively).

**Figure 13 sensors-24-05909-f013:**
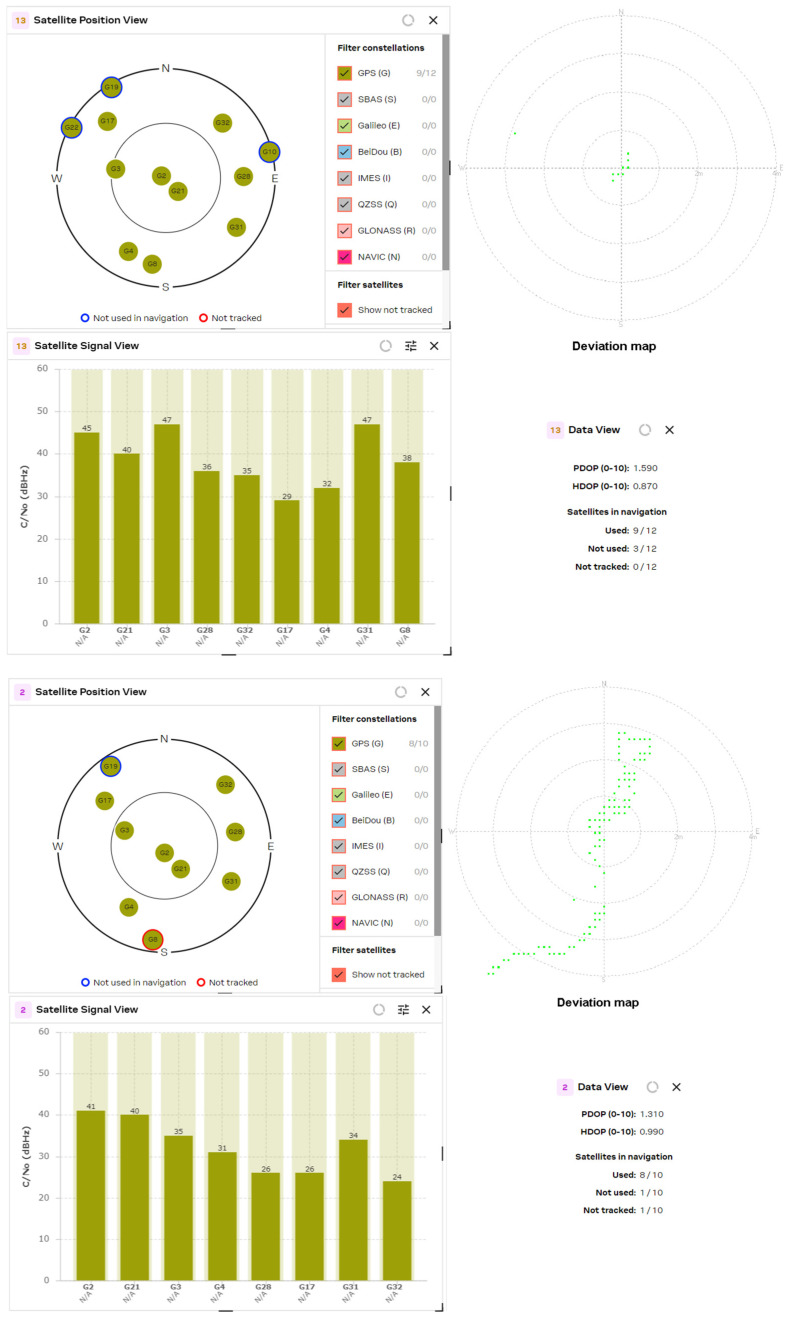
MinewSemi MS32SN1 receiver (patch and helix antenna, respectively).

**Figure 14 sensors-24-05909-f014:**
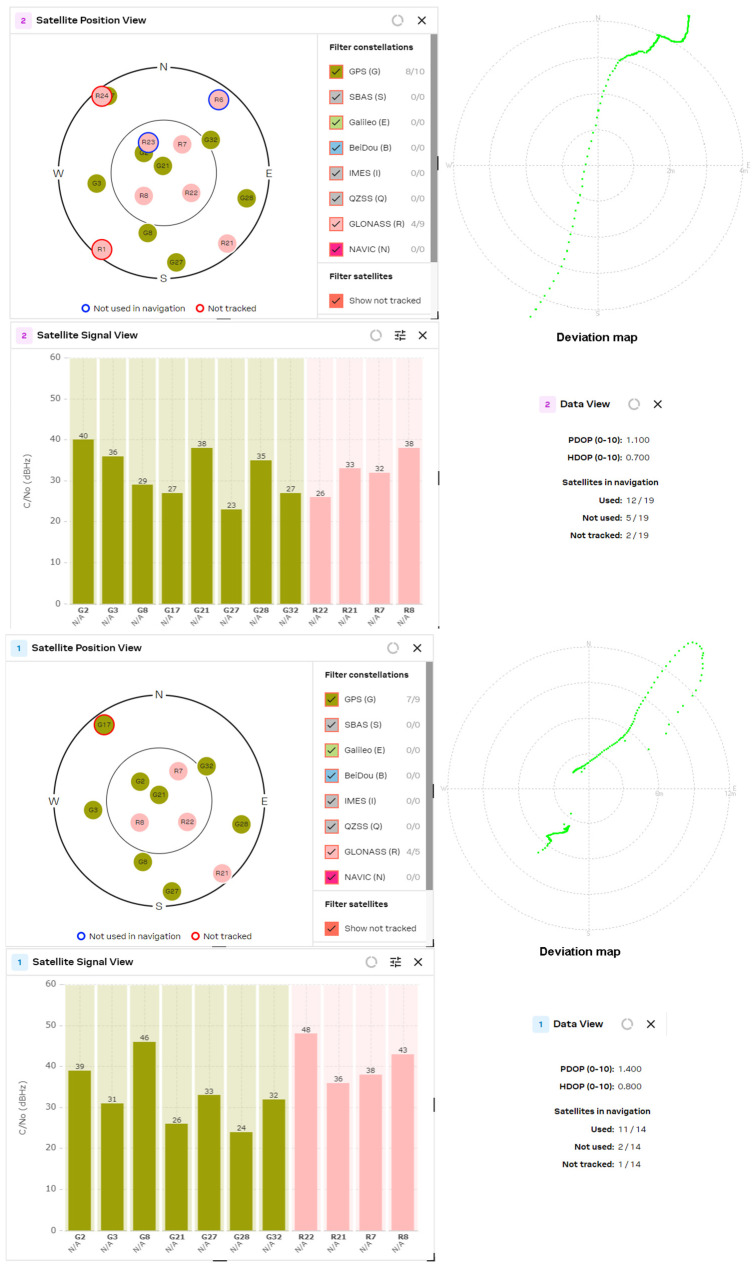
ATGM336H receiver (patch and helix antenna, respectively).

**Figure 15 sensors-24-05909-f015:**
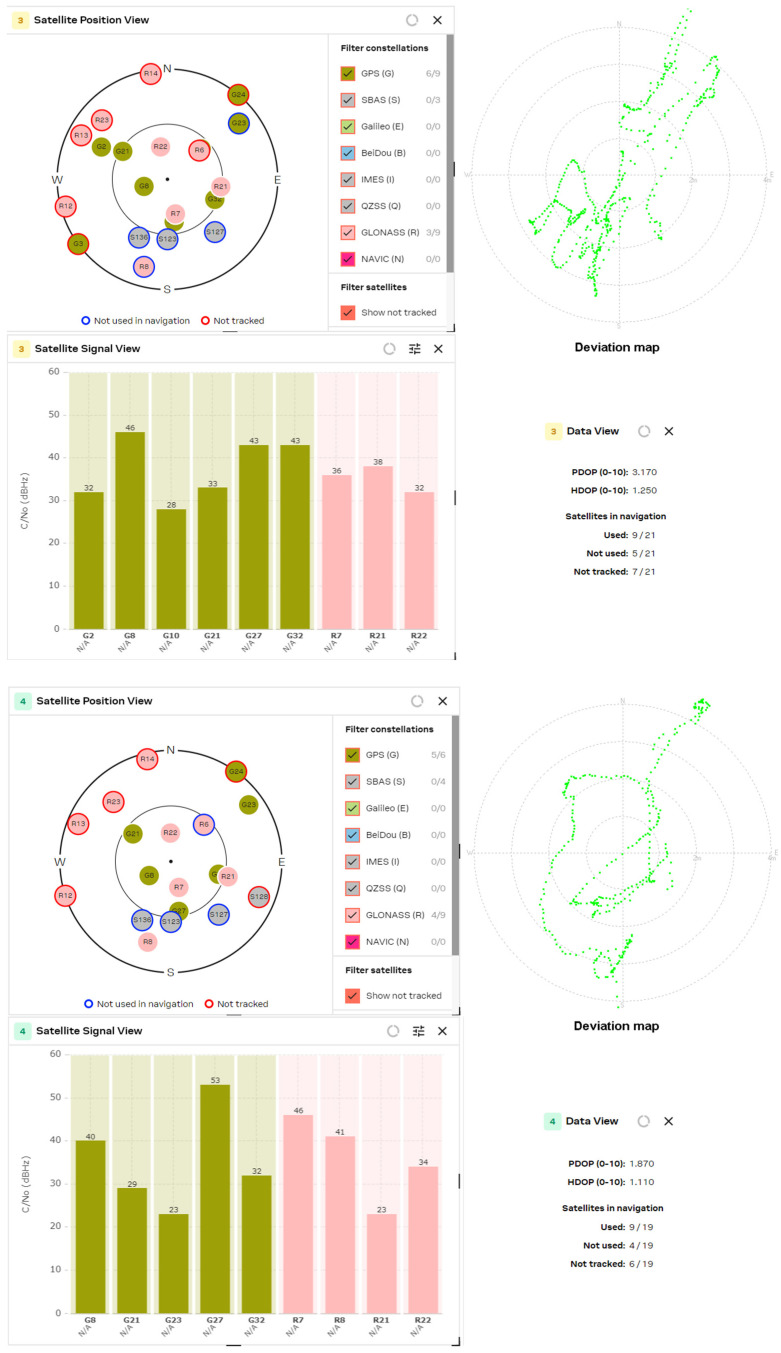
U-blox MAX M8Q (patch and helix antenna, respectively).

**Figure 16 sensors-24-05909-f016:**
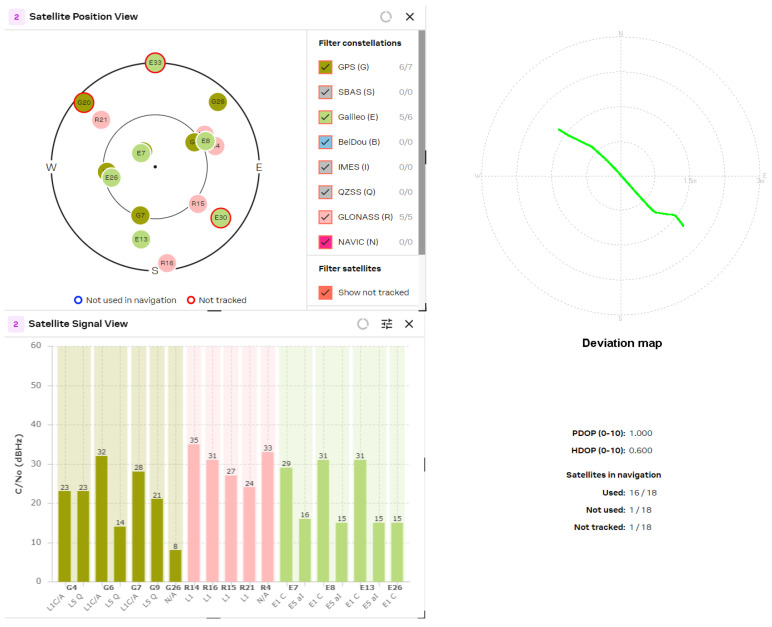
MinewSemi ME32GR01 receiver (patch and helix antenna, respectively).

**Figure 17 sensors-24-05909-f017:**
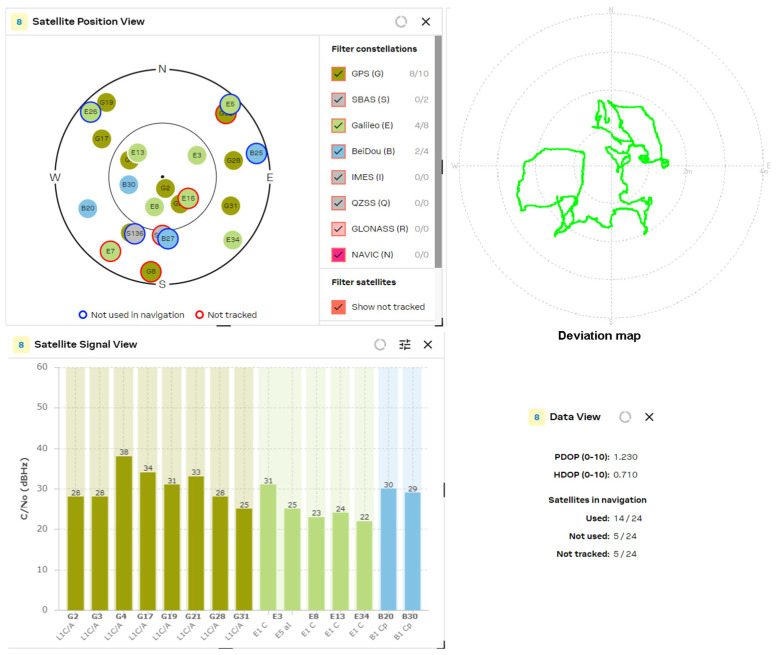
U-blox NEO F10T receiver (patch and helix antenna, respectively).

**Table 1 sensors-24-05909-t001:** Frequency bands and GNSS signal description.

Frequency Band	PRN Code		Code Length [Chip]	Code Baud Rate [Mcps]	Modulation Scheme	Bandwidth [MHz]	Data Baud Rate [sps/bps]
L1	C/A		1023	1.023	BPSK ^1^ (1)	2.046	50/50
	P		~7 days	10.23	BPSK (10)	20.46	50/50
	M		-	5.115	BOC ^2^ (10.5)	30.69	-
	L1C_D_ ^3^		10,230	1.023	BOC (1.1)	4.092	100/50
	L1C_p_ ^3^		102,301,800	1.023	BOC (1.1)	4.092	-
L2	P		~7 days	10.23	BPSK (10)	20.46	50/50
	L2C	M	10,230	1.023	BPSK (1)	2.046	50/25
		L	767,250				-
	M			5.115	BOC (10.5)	30.69	-
L5	L5I ^4^		10,230.10	10.23	BPSK (10)	20.46	100/50
	L5Q ^4^		10,230.20	10.23	BPSK (10)	20.46	-

^1^ BPSK (*n*)—Binary Phased Shift Keying with chip rate equal to *n* × 1.023 MHz, ^2^ BOC (*n*,*m*)—Binary Offset Carrier modulation with *n* being a 1.023 MHz square wave and *m* being a 1.023 MHz pseudo-random code [14], ^3^ Overlay and ranging codes for GPS L1C code [15], ^4^ L5 PRN codes [16].

## Data Availability

Data are contained within the article.

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
