# Peer review of "Study of Global Navigation Satellite System Receivers’ Accuracy for Unmanned Vehicles"

_sensors, 2024, doi:10.3390/s24185909_

Round 1

Reviewer 1 Report

Comments and Suggestions for Authors

This paper investigates high precision navigation for unmanned vehicles. My comments are provided as follows:

1. In abstract, both “dual-band GNSS receivers” and “dual-frequency GNSS receivers” are used. Please clarify the difference between “dual-band” and “dual-frequency”.

2. “Figure.1.”, “Figure.2.”, “Figure.3.”, “Figure. 4.” and “Figure. 5” should be “Figure 1.”, “Figure 2.”, “Figure 3.”, “Figure 4.”, and “Figure 5.”, respectively.

3. The GNSS receiver has been widely investigated in literature. The main contributions of obtaining some test results are not enough. In my opinion, new contributions should be provided.

4. The legends of Fig. 7, Fig. 9, and Fig. 10 should be improved. The current legends are a bit messy and not shown clearly.

5. High resolution images should be provided for Figs. 16-21.

6. There are too few references. The latest relevant references should be provided.

Author Response

Comments and Suggestions for Authors

This paper investigates high precision navigation for unmanned vehicles. My comments are provided as follows:

*1. In abstract, both “dual-band GNSS receivers” and “dual-frequency GNSS receivers” are used. Please clarify the difference between “dual-band” and “dual-frequency”.
-Remark accepted. The terminology is unified to "dual-band"

*2. “Figure.1.”, “Figure.2.”, “Figure.3.”, “Figure. 4.” and “Figure. 5” should be “Figure 1.”, “Figure 2.”, “Figure 3.”, “Figure 4.”, and “Figure 5.”, respectively.
-Remark accepted. The figure titles are corrected.

*3. The GNSS receiver has been widely investigated in literature. The main contributions of obtaining some test results are not enough. In my opinion, new contributions should be provided.
-Remark accepted. The study of the GNSS receivers' accuracy is provided towards the GNSS applications to correct the drawbacks of the IMU systems. The paragraph is added to the introduction to specify the investigation goal (lines 108-133).

*4. The legends of Fig. 7, Fig. 9, and Fig. 10 should be improved. The current legends are a bit messy and not shown clearly.
-Remark accepted. The figures are made as clear as possible. The legends of the above mentioned figures are corrected and improved.

*5. High resolution images should be provided for Figs. 16-21.
- Remark accepted. The figures quality is defined by u-center software. The text is well readable

*6. There are too few references. The latest relevant references should be provided.
- Remark accepted. After the revision of the paper and provision of additional analysis, discussions and data, the number of references significantly increased.

* Asterisk marks the Reviewer's comments
- Dash marks the Author's response

Reviewer 2 Report

Comments and Suggestions for Authors

This manuscript aims to evaluate the performance of dual-frequency GNSS receiver combined with inertial sensors for high-precision navigation tasks in the context of unmanned vehicles. The paper presents a comparison study of several GNSS modules using different antennas and analyze their accuracy in various conditions. While the study addresses an important and interesting topic, the manuscript suffers from several critical weaknesses, including poor organization, lack of clarity in presenting results, and insufficient discussion of the broader implications of the findings. I think this manuscript is not acceptable. And in my opinion, I recommend rejection. If the authors insist on resubmission, before the manuscript can be considered for publication, the following comments, although not complete, must be addressed.

[Title]

Line 2: this is not a very good title for a research paper. It is broad. It would be good if the title is more specific to reflect the core study in this paper.

[Abstract]

Line 9-25: the abstract is quite verbose, and it lacks focus. The current form makes it difficult to grasp the study’s novelty. The key contributions and findings of this study should be clearly stated and highlighted. Besides, there are quite a few abbreviations and technical jargons in the abstract. The full wording and/or necessary description should be given for the first time when they appear in a paper. What do you mean by “electronic production”? Advances in electronic production alone can increase the number of the tracking satellites? I don’t think so.    

[Introduction]

Line 28-38: the introduction seems to lack a critical review of existing literature. While few references are provides, they are not adequately discussed to highlight the research gaps this study aims to address.

Line 39-96: Again, there are few abbreviations and/or technical jargons that should be explained at the first place. Besides, the objectives of this study are not clearly stated, making it challenging to understand the purpose of the experiments. Please provide your objectives clearly and also provide the contributions of this study clearly at the end of the introduction.

Figure 1: why the figure is titled as “new” signal…? Why “new”?

Table 1: what is L2CD? What is L1CP? What is L1I? What is L1Q? What is BPSK? What is BOC?

[Precise point positioning (PPP) with dual frequency GNSS navigation]

Line 97-198: looking at the section title, my first impression is that this is another section of introduction. There should clearly be a section to provide the methodology to address the objectives in this study. There are no objectives anyway in the introduction. This is not good. Besides, this section is overly technical without sufficient explanation of the rationale behind certain design choices. Why are those antennas chosen? What is NEO? What is MAX? What is u-blox? What is NEO-M9N? What is GP-01? …. Basically, the reasoning for selecting specific GNSS modules should be more clearly justifies. Why an USB connection so important to mention here?

Line 122-137: the helix antenna design section could benefit from a more detailed explanation of how the design parameters are chosen and their expected impact on performance.

Figure 15-21: Those figures are not corresponding in the text. And there are no explanations.

Line 242-270: The discussion of the HDOP and PDOP values is superficial and does not fully explore their implications for the accuracy and reliability of the GNSS receivers.

[Discussion]

Line 271-297: the discussion section does not critically engage with the results or compare them with existing studies. There is a lack of depth in analyzing how the findings contribute to the field of high-precision navigation.

Line 302-313: the limitations of the study are not adequately addressed, particularly concerning the applicability of the findings to real-world scenarios

[References]

Line 321-344: only 11 references?

Author Response

This manuscript aims to evaluate the performance of dual-frequency GNSS receiver combined with inertial sensors for high-precision navigation tasks in the context of unmanned vehicles. The paper presents a comparison study of several GNSS modules using different antennas and analyze their accuracy in various conditions. While the study addresses an important and interesting topic, the manuscript suffers from several critical weaknesses, including poor organization, lack of clarity in presenting results, and insufficient discussion of the broader implications of the findings. I think this manuscript is not acceptable. And in my opinion, I recommend rejection. If the authors insist on resubmission, before the manuscript can be considered for publication, the following comments, although not complete, must be addressed.

*Line 2: this is not a very good title for a research paper. It is broad. It would be good if the title is more specific to reflect the core study in this paper.
- Remark accepted. The paper title is change to narrow the field of the study. 

[Abstract]
*Line 9-25: the abstract is quite verbose, and it lacks focus. The current form makes it difficult to grasp the study’s novelty. The key contributions and findings of this study should be clearly stated and highlighted. Besides, there are quite a few abbreviations and technical jargons in the abstract. The full wording and/or necessary description should be given for the first time when they appear in a paper. What do you mean by “electronic production”? Advances in electronic production alone can increase the number of the tracking satellites? I don’t think so.    
-Remark accepted. The abstract is revised to change or remove some jargons, abbreviations and inaccuracies such as "electronic production". Under "electronic production" the authors had in mind the advances in the performance  (in terms of speed, memory and computational power with each new generation) of the integrated CPU in the GNSS receivers, which allows to increase the number of correlations, i.e. the number of simultaneously tracked satellites. That is now reworded, in order to avoid ambiguity. On other hand, following the Reviewers comment, the Introduction part of the paper is reworked, to clearly state and define the purpose of the study.

[Introduction]
*Line 28-38: the introduction seems to lack a critical review of existing literature. While few references are provides, they are not adequately discussed to highlight the research gaps this study aims to address.
-Remark accepted. Additional references are added in the introduction to describe the state of the art of the problems in the position calculations. At the end of the introduction the paper objectives are described.

*Line 39-96: Again, there are few abbreviations and/or technical jargons that should be explained at the first place. Besides, the objectives of this study are not clearly stated, making it challenging to understand the purpose of the experiments. Please provide your objectives clearly and also provide the contributions of this study clearly at the end of the introduction.
- Remark accepted. The objectives are added in the introduction and the experiments purpose is now more clearly defined.

*Figure 1: why the figure is titled as “new” signal…? Why “new”?
- Remark accepted. The figure title is corrected.

*Table 1: what is L2CD? What is L1CP? What is L1I? What is L1Q? What is BPSK? What is BOC?
- Remark accepted under condition. The abbreviations are well popular in GNSS community, however the reviewer is correct and to make the paper  easy readable for a broader auditory, description of the abbreviations is added below Table 1 and at the end of the paper.

*[Precise point positioning (PPP) with dual frequency GNSS navigation]
Line 97-198: looking at the section title, my first impression is that this is another section of introduction. There should clearly be a section to provide the methodology to address the objectives in this study. There are no objectives anyway in the introduction. This is not good. Besides, this section is overly technical without sufficient explanation of the rationale behind certain design choices. Why are those antennas chosen? What is NEO? What is MAX? What is u-blox? What is NEO-M9N? What is GP-01? …. Basically, the reasoning for selecting specific GNSS modules should be more clearly justifies. Why an USB connection so important to mention here?
- Remark accepted under condition. The abbreviation description and references are added. Since the authors select the latest generation GNSS receivers, even their vendors and developers use some "engineering samples" to provide the positioning experiments. In the revised version of the paper, a new section is added (lines 108-133) to describe the technical rationale, as requested by the Reviewer. The choice of the antennas is made on the basis of the bandwidth, gain, radiation pattern and system application. It is described in the updated version of the paper, lines 164-268.

* Line 122-137: the helix antenna design section could benefit from a more detailed explanation of how the design parameters are chosen and their expected impact on performance.
- Remark accepted. The helix antenna design parameters are chosen according to its application at GNSS systems, i.e. to cover the bandwidth requirement and phased center position. The results are compared with other published antennas. A detailed paragraph is added (lines 164-268).

* Figure 15-21: Those figures are not corresponding in the text. And there are no explanations.
- Remark accepted. The description and the analysis of the above mentioned figures is added in the section Discussion. 

*Line 242-270: The discussion of the HDOP and PDOP values is superficial and does not fully explore their implications for the accuracy and reliability of the GNSS receivers.
- Remark accepted under condition. DOP parameters are widely used in the literature to evaluate the position accuracy. The authors mentioned only the variables which influence over their valies.

[Discussion]
*Line 271-297: the discussion section does not critically engage with the results or compare them with existing studies. There is a lack of depth in analyzing how the findings contribute to the field of high-precision navigation.
- Remark accepted. The discussion is significantly enlarged with figure comments and comparison results in the another literature sources.

*Line 302-313: the limitations of the study are not adequately addressed, particularly concerning the applicability of the findings to real-world scenarios
-Remark accepted. The application field are added at the end of the conclusion. The results may be used not only for standalone GNSS systems but also for IMU/GNSS systems to set the parameters of the Kalman filters

[References]
*Line 321-344: only 11 references?
Remark accepted . The number of references is significantly increased to improve the analysis. Additional references are added to the revised introduction and discussion, to describe the objectives in details and to compare the obtained results with the existing studies.

* Asterisk marks the Reviewer's comments
- Dash marks the Author's response

Round 2

Reviewer 1 Report

Comments and Suggestions for Authors

The authors have replied to my previous comments. I have no more comments.

Author Response

Thank you for the valuable comments in the Round1 !
They really helped us to improve the manuscript

Reviewer 2 Report

Comments and Suggestions for Authors

It seems that the authors have made efforts to improve. However, this is still a poor written manuscript in my standard. IT IS VERY DIFFICULT TO FOLLOW WHAT THE AUTHORS ARE TRYING TO DO. And how the conclusions are supported by the experiment and analysis is not clear. For example for this conclusion "he integration of GNSS receivers with an inertial measurement unit (IMU) may overcome this navigation problem for a limited period of time as IMU systems may calculate the spatial object position and its speed and movement direction, but the position calculation is a relative process, which leads to the unlimited error accumulation, but the position refresh rate may exceed 400Hz. The integration of both navigation systems may significantly reduce their drawbacks.", it is not clear how you arrived at it.  And what is the sign between the two numbers in "0.9÷1.0"?  

Author Response

Comment1:  IT IS VERY DIFFICULT TO FOLLOW WHAT THE AUTHORS ARE TRYING TO DO. And how the conclusions are supported by the experiment and analysis is not clear. For example for this conclusion the integration of GNSS receivers with an inertial measurement unit (IMU) may overcome this navigation problem for a limited period of time as IMU systems may calculate the spatial object position and its speed and movement direction, but the position calculation is a relative process, which leads to the unlimited error accumulation, but the position refresh rate may exceed 400Hz. The integration of both navigation systems may significantly reduce their drawbacks.", it is not clear how you arrived at it.

Answer1:
The paper discusses comprehensive analysis of  Dilution of Precision in several GPS receiver arrangements. Since DOP parameters are directly involved in the Kalman filter matrices and the receiver realization is hardware dependent,  the paper delivers the most actual status and capabilities of the hardware, that is not discussed so far in other publications, to the best of our knowledge and will be of great interest of the relevant auditory. It is presented in a clear way, in separate chapters starting from methodology, the hardware and material description, the approach, the experimental results and the conclusion. The study reflects the current state of the art of the technology and has a direct applicability for current design and future studies. 

Comment 2  "The integration of GNSS receivers with an inertial measurement unit (IMU) may overcome this navigation problem for a limited period of time as IMU systems may calculate the spatial object position and its speed and movement direction, but the position calculation is a relative process, which leads to the unlimited error accumulation, but the position refresh rate may exceed 400Hz. The integration of both navigation systems may significantly reduce their drawbacks.", it is not clear how you arrived at it“  

Answer 2: It is an essential, well known drawback of the Inertial navigation systems, that they accumulate error with time, and this error becomes substantial even within a very short amount of time. The natural solution of this problem is  augmentation of the Inertial system with GNSS, where the GNSS will correct as time goes and Inertial will supplement navigation data, where the GNSS signal is unavailable, jammed  or suppressed due to noise. The paper clearly demonstrates the benefits of such augmentation and the expected results in the current state of the art. 

Comment 3: W
hat is the sign between the two numbers in "0.9÷1.0” 

Answer 3: This is a notation for range. We agree that it may be not widespread and replaced is with more popular one - brackets [0.9, 1]